# Active Learning from Peers

**Keerthiram Murugesan**  **Jaime Carbonell**
School of Computer Science
Carnegie Mellon University
Pittsburgh, PA 15213
{kmuruges,jgc}@cs.cmu.edu

## Abstract

This paper addresses the challenge of learning from peers in an online multitask setting. Instead of always requesting a label from a human oracle, the proposed method first determines if the learner for each task can acquire that label with sufficient confidence from its peers either as a task-similarity weighted sum, or from the single most similar task. If so, it saves the oracle query for later use in more difficult cases, and if not it queries the human oracle. The paper develops the new algorithm to exhibit this behavior and proves a theoretical mistake bound for the method compared to the best linear predictor in hindsight. Experiments over three multitask learning benchmark datasets show clearly superior performance over baselines such as assuming task independence, learning only from the oracle and not learning from peer tasks.

## 1  Introduction

Multitask learning leverages the relationship between the tasks to transfer relevant knowledge from information-rich tasks to information-poor ones. Most existing work in multitask learning focuses on how to take advantage of these task relationships, either by sharing data directly [1] or learning model parameters via cross-task regularization techniques [2, 3, 4]. This paper focuses on a specific multitask setting where tasks are allowed to interact by requesting labels from other tasks for difficult cases.

In a broad sense, there are two settings to learn multiple related tasks together: 1) batch learning, in which an entire training set is available to the learner 2) online learning, in which the learner sees the data sequentially. In recent years, online multitask learning has attracted increasing attention [5, 6, 7, 8, 9, 10]. The online multitask setting starts with a learner at each round $t$, *receiving* an example (along with a task identifier) and *predicts* the output label. One may also consider learning multiple tasks simultaneously by receiving $K$ examples for $K$ tasks at each round $t$. Subsequently, the learner receives the true label and *updates* the model(s) as necessary. This sequence is repeated over the entire data, simulating a data stream. In this setting, the assumption is that the true label is readily available for the task learner, which is impractical in many applications.

Recent works in active learning for sequential problems have addressed this concern by allowing the learner to make a decision on whether to ask the oracle to provide the true label for the current example and incur a cost or to skip this example. Most approach in active learning for sequential problems use a measure such a confidence of the learner in the current example [11, 12, 13, 14, 15]. In online multitask learning, one can utilize the task relationship to further reduce the total number of labels requested from the oracle. This paper presents a novel active learning for the sequential decision problems using *peers* or *related tasks*. The key idea is that when the learner is not confident on the current example, the learner is allowed to query its peers, which usually has a low cost, before requesting a true label from the oracle and incur a high cost. Our approach follows a perceptron-based update rule in which the model for a given task is updated only when the prediction for that task is

in error. The goal of an online learner in this setting is to minimize errors attempting to reach the performance of the full hindsight learner and at the same time, reduce the total number of queries issued to the oracle.

There are many useful application areas for online multitask learning with selective sampling, including optimizing financial trading, email prioritization and filtering, personalized news, crowd source-based annotation, spam filtering and spoken dialog system, etc. Consider the latter, where several automated agents/bots servicing several clients. Each agent is specialized or trained to answer questions from customers on a specific subject such as automated payment, troubleshooting, adding or cancelling services, etc. In such setting, when one of the automated agents cannot answer a customer's question, it may request the assistance of another automated agent that is an expert in the subject related to that question. For example, an automated agent for customer retention may request some help from an automated agent for new services to offer new deals for the customer. When both the agents could not answer the customer's question, the system may then direct the call to a live agent. This may reduce the number of service calls directed to live agents and the cost associated with such requests.

Similarly in spam filtering, where some spam is universal to all users (e.g. financial scams), some messages might be useful to certain affinity groups, but spam to most others (e.g. announcements of meditation classes or other special interest activities), and some may depend on evolving user interests. In spam filtering each user is a task, and shared interests and dis-interests formulate the inter-task relationship matrix. If we can learn the task relationship matrix as well as improving models from specific decisions from peers on difficult examples, we can perform mass customization of spam filtering, borrowing from spam/not-spam feedback from users with similar preferences. The primary contribution of this paper is precisely active learning for multiple related tasks and its use in estimating per-task model parameters in an online setting.

## 1.1 Related Work

While there is considerable literature in online multitask learning, many crucial aspects remain largely unexplored. Most existing work in online multitask learning focuses on how to take advantage of task relationships. To achieve this, Lugosi et. al [7] imposed a hard constraint on the $K$ simultaneous actions taken by the learner in the expert setting, Agarwal et. al [16] used matrix regularization, and Dekel et. al [6] proposed a global loss function, as an absolute norm, to tie together the loss values of the individual tasks. In all these works, their proposed algorithms assume that the true labels are available for each instance.

Selective sampling-based learners in online setting, on the other hand, decides whether to ask the human oracle for labeling of difficult instances [11, 12, 13, 14, 15]. It can be easily extended to online multitask learning setting by applying selective sampling for each individual task separately. Saha et. al [9] formulated the learning of task relationship matrix as a Bregman-divergence minimization problem w.r.t. positive definite matrices and used this task-relationship matrix to naively select the instances for labelling from the human oracle.

Several recent works in online multitask learning recommended updating all the task learners on each round $t$ [10, 9, 8]. When a task learner makes a mistake on an example, all the tasks' model parameters are updated to account for the new examples. This significantly increases the computational complexity at each round, especially when the number of tasks is large [17]. Our proposed method avoids this issue by updating only the learner of the current example and utilize the knowledge from peers only when the current learner requested them.

This work is motivated by the recent interests in active learning from multiple (strong or weak) teachers [18, 19, 12, 20, 21, 22]. Instead of single all-known oracle, these papers assume multiple oracles (or teachers) each with a different area of expertise. At round $t$, some of the teachers are experts in the current instance but the others may not be confident in their predicted labels. Such learning setting is very common in crowd-sourcing platform where multiple annotators are used to label an instance. Our learning setting is different from their approaches where, instead of learning from multiple oracles, we learn from our peers (or related tasks) without any associated high cost. Finally, our proposed method is closely related to learning with rejection option [23, 24] where the learner may choose not to predict label for an instance. To reject an instance, they use a measure of

1. Receive an example $x^{(t)}$ for the task $k$
2. If the task $k$ is not confident in the prediction for this example, ask the *peers* or *related tasks* whether they can give a confident label to this example.
3. If the *peers* are not confident enough, ask the oracle for the true label $y^{(t)}$.

Figure 1: Proposed learning approach from peers.

confidence to identify difficult instances. We use a similar approach to identify when to query peers and when to query the human oracle for true label.

## 2 Problem Setup

Suppose we are given $K$ tasks where the $k^{th}$ task is associated with $N_k$ training examples. For brevity, we consider a binary classification problem for each task, but the methods generalize to multi-class settings and are also applicable to regression tasks. We denote by $[N]$ consecutive integers ranging from 1 to $N$. Let $\{(x_k^{(i)}, y_k^{(i)})\}_{i=1}^{N_k}$ be data for task $k$ where $x_k^{(i)} \in \mathbb{R}^d$ is the $i^{th}$ instance from the $k^{th}$ task and $y_k^{(i)}$ is its corresponding true label. When the notation is clear from the context, we drop the index $k$ and write $((x^{(i)}, k), y^{(i)})$.

Let $\{w_k^*\}_{k \in [K]}$ be any set of arbitrary vectors where $w_k^* \in \mathbb{R}^d$. The hinge losses on the example $((x^{(t)}, k), y^{(t)})$ are given by $\ell_{kk}^{(t)*} = \left(1 - y^{(t)} \langle x^{(t)}, w_k^* \rangle\right)_+$ and $\ell_{km}^{(t)*} = \left(1 - y^{(t)} \langle x^{(t)}, w_m^* \rangle\right)_+$, respectively, where $(z)_+ = \max(0, z)$. Similarly, we define hinge losses $\ell_{kk}^{(t)}$ and $\ell_{km}^{(t)}$ for the linear predictors $\{w_k^{(t)}\}_{k \in [K]}$ learned at round $t$. Let $Z^{(t)}$ be a Bernoulli random variable to indicate whether the learner requested a true label for the example $x^{(t)}$. Let $M^{(t)}$ be a binary variable to indicate whether the learner made a mistake on the example $x^{(t)}$. We use the following expected hinge losses for our theoretical analysis: $\tilde{L}_{kk} = \mathbb{E}\left[\sum_t M^{(t)} Z^{(t)} \ell_{kk}^{(t)*}\right]$ and $\tilde{L}_{km} = \mathbb{E}\left[\sum_t M^{(t)} Z^{(t)} \ell_{km}^{(t)*}\right]$.

We start with our proposed active learning from peers algorithm based on selective sampling for online multitask problems and study the mistake bound for the algorithm in Section 3. We report our experimental results and analysis in Section 4. Additionally, we extend our learning algorithm to learning multiple task in parallel in the supplementary.

## 3 Learning from Peers

We consider multitask perceptron for our online learning algorithm. On each round $t$, we receive an example $(x^{(t)}, k)$ from task $k$ [1]. Each perceptron learner for the task $k$ maintains a model represented by $w_k^{(t-1)}$ learned from examples received until round $t-1$. Task $k$ predicts a label for the received example $x^{(t)}$ using $h_k(x^{(t)}) = \langle w_k^{(t-1)}, x^{(t)} \rangle$ [2]. As in the previous works [11, 12, 23], we use $|h_k(x^{(t)})|$ to measure the confidence of the $k^{th}$ task learner on this example. When the confidence is higher, the learner doesn't require the need to request the true label $y^{(t)}$ from the oracle.

Built on this idea, [11] proposed a selective sampling algorithm using the margin $|h_k(x^{(t)})|$ to decide whether to query the oracle or not. Intuitively, if $|h_k(x^{(t)})|$ is small, then the $k^{th}$ task learner is not confident in the prediction of $x^{(t)}$ and vice versa. They consider a Bernoulli random variable $P^{(t)}$ for the event $|h_k(x^{(t)})| \le b_1$ with probability $\frac{b_1}{b_1 + |h_k(x^{(t)})|}$ for some predefined constant $b_1 \ge 0$. If

$P^{(t)} = 1$ (confidence is low), then the $k^{th}$ learner requests the oracle for the true label. Similarly when $P^{(t)} = 0$ (confidence is high), the learner skips the request to the oracle. This considerably saves a lot of label requests from the oracle. When dealing with multiple tasks, one may use similar idea and apply selective sampling for each task individually [25]. Unfortunately, such approach doesn't take into account the inherent relationship between the tasks.

In this paper, we consider a novel active learning (or selective sampling) for online multitask learning to address the concerns discussed above. Our proposed learning approach can be summarized in Figure 1. Unlike in the previous work [8, 9, 10], we update only the current task parameter $w_k$ when we made a mistake at round $t$, instead of updating all the task model parameters $w_m, \forall m \in [K], m \neq k$. Our proposed method avoids this issue by updating only the learner of the current example and share the knowledge from peers only when the assistance is needed. In addition, the task relationship is taken into account, to measure whether the peers are confident in predicting this example. This approach provides a compromise between learning them independently and learning them by updating all the learners when a specific learner makes a mistake.

As in traditional selective sampling algorithm [11], we consider a Bernoulli random variable $P^{(t)}$ for the event $|h_k(x^{(t)})| \leq b_1$ with probability $\frac{b_1}{b_1 + |h_k(x^{(t)})|}$. In addition, we consider a second Bernoulli random variable $Q^{(t)}$ for the event $|h_m(x^{(t)})| \leq b_2$ with probability $\frac{b_2}{b_2 + \sum_{m \in [K], m \neq k} \tau_{km}^{(t-1)} |h_m(x^{(t)})|}$.
The idea is that when the weighted sum of the confidence of the peers on the current example is high, then we use the predicted label $\tilde{y}^{(t)}$ from the peers for the perceptron update instead of requesting a true label $y^{(t)}$ from the oracle. In our experiment in Section 4, we consider the confidence of most related task instead of the weighted sum to reduce the computational complexity at each round. We set $Z^{(t)} = P^{(t)} Q^{(t)}$ and set $M^{(t)} = 1$ if we made a mistake at round $t$ i.e., $(y^{(t)} \neq \hat{y}^{(t)})$ (only when the label is revealed/queried).

The pseudo-code is in Algorithm 1. Line 14 is executed when we request a label from the oracle or when peers are confident on the label for the current example. Note the two terms in $(M^{(t)} Z^{(t)} y^{(t)} + \tilde{Z}^{(t)} \tilde{y}^{(t)})$ are mutually exclusive (when $P^{(t)} = 1$). Line (15-16) computes the relationship between tasks $\tau_{km}$ based on the recent work by [10]. It maintains a distribution over peers w.r.t the current task. The value of $\tau$ is updated at each round using the cross-task error $\ell_{km}$. In addition, we use the $\tau$ to get the confidence of the most-related task rather than the weighted sum of the confidence of the peers to get the predicted label from the peers (see Section 4 for more details). When we are learning with many tasks [17], it provides a faster computation without significantly compromising the performance of the learning algorithm. One may use different notion of task relationship based on the application at hand. Now, we give the bound on the expected number of mistakes.

**Theorem 1.** *let $S_k = \left\{ \left( (x^{(t)}, k), y^{(t)} \right) \right\}_{t=1}^{T}$ be a sequence of $T$ examples given to Algorithm 1 where $x^{(t)} \in \mathbb{R}^d$, $y^{(t)} \in \{-1, +1\}$ and $X = \max_t \|x^{(t)}\|$. Let $P^{(t)}$ be a Bernoulli random variable for the event $|h_k(x^{(t)})| \leq b_1$ with probability $\frac{b_1}{b_1 + |h_k(x^{(t)})|}$ and let $Q^{(t)}$ be a Bernoulli random variable for the event $|h_m(x^{(t)})| \leq b_2$ with probability $\frac{b_2}{b_2 + \max_{\substack{m \in [K] \\ m \neq k}} |h_m(x^{(t)})|}$. Let $Z^{(t)} = P^{(t)} Q^{(t)}$ and $M^{(t)} = \mathbb{I}(y^{(t)} \neq \hat{y}^{(t)})$.*

*If the Algorithm 1 is run with $b_1 > 0$ and $b_2 > 0$ ($b_2 \geq b_1$), then $\forall t \geq 1$ and $\gamma > 0$ we have*

$$\mathbb{E}\left[ \sum_{t \in [T]} M^{(t)} \right] \leq \frac{b_2}{\gamma} \left[ \frac{(2b_1 + X^2)^2}{8b_1 \gamma} \left( \|w_k^*\|^2 + \max_{m \in [K], m \neq k} \|w_m^*\|^2 \right) \right.$$

$$\left. + \left( 1 + \frac{X^2}{2b_1} \right) \left( \tilde{L}_{kk} + \max_{m \in [K], m \neq k} \tilde{L}_{km} \right) \right]$$

*Then, the expected number of label requests to the oracle by the algorithm is*

$$\sum_t \frac{b_1}{b_1 + |h_k(x^{(t)})|} \frac{b_2}{b_2 + \max_{\substack{m \in [K] \\ m \neq k}} |h_m(x^{(t)})|}$$

**Algorithm 1:** Active Learning from Peers

**Input :** $b_1 > 0$, $b_2 > 0$ s.t., $b_2 \geq b_1$, $\lambda > 0$, Number of rounds $T$

1    *Initialize* $w_m^{(0)} = \mathbf{0} \; \forall m \in [K]$, $\boldsymbol{\tau}^{(0)}$.

2  **for** $t = 1 \ldots T$ **do**

3      *Receive* $(x^{(t)}, k)$

4      *Compute* $\hat{p}_{kk}^{(t)} = \langle x^{(t)}, w_k^{(t-1)} \rangle$

5      *Predict* $\hat{y}^{(t)} = sign(\hat{p}_{kk}^{(t)})$

6      *Draw* a Bernoulli random variable $P^{(t)}$ with probability $\frac{b_1}{b_1 + |\hat{p}_{kk}^{(t)}|}$

7      **if** $P^{(t)} = 1$ **then**

8         *Compute* $\hat{p}_{km}^{(t)} = \langle x^{(t)}, w_m^{(t-1)} \rangle \; \forall m \neq k, m \in [K]$

9         *Compute* $\tilde{p}^{(t)} = \sum_{m \neq k, m \in [K]} \tau_{km}^{(t-1)} \hat{p}_{km}^{(t)}$ and $\tilde{y}^{(t)} = sign(\tilde{p}^{(t)})$

10        *Draw* a Bernoulli random variable $Q^{(t)}$ with probability $\frac{b_2}{b_2 + |\tilde{p}^{(t)}|}$

11      **end**

12      Set $Z^{(t)} = P^{(t)} Q^{(t)}$ & $\tilde{Z}^{(t)} = P^{(t)}(1 - Q^{(t)})$

13      *Query* true label $y^{(t)}$ if $Z^{(t)} = 1$ and set $M^{(t)} = 1$ if $\hat{y}^{(t)} \neq y^{(t)}$

14      *Update* $w_k^{(t)} = w_k^{(t-1)} + (M^{(t)} Z^{(t)} y^{(t)} + \tilde{Z}^{(t)} \tilde{y}^{(t)}) x^{(t)}$

15      *Update* $\tau$:

16

$$\tau_{km}^{(t)} = \frac{\tau_{km}^{(t-1)} e^{-\frac{Z^{(t)}}{\lambda} \ell_{km}^{(t)}}}{\sum_{\substack{m' \in [K] \\ m' \neq k}} \tau_{km'}^{(t-1)} e^{-\frac{Z^{(t)}}{\lambda} \ell_{km'}^{(t)}}} \quad m \in [K], m \neq k \qquad (1)$$

17 **end**

The proof is given in Appendix A. It follows from Theorem 1 in [11] and Theorem 1 in [10] and setting $b_2 = b_1 + \frac{X^2}{2} + \frac{\|w_k^*\|}{2}$, where $b_1 = \frac{X^2}{2}\sqrt{1 + \frac{4\gamma^2}{\|w_k^*\|X^2}\frac{\tilde{L}_{kk}}{\gamma}}$. Theorem 1 states that the quality of the bound depends on both $\tilde{L}_{kk}$ and the maximum of $\{\tilde{L}_{km}\}_{m \in [K], m \neq k}$. In other words, the worst-case regret will be lower if the $k^{th}$ true hypothesis $w_k^*$ can predict the labels for training examples in both the $k^{th}$ task itself as well as those in all the other related tasks in high confidence. In addition, we consider a related problem setting in which all the $K$ tasks receive an example simultaneously. We give the learning algorithm and mistake bound for this setting in Appendix B.

## 4 Experiments

We evaluate the performance of our algorithm in the online setting. All reported results in this section are averaged over 10 random runs on permutations of the training data. We set the value of $b_1 = 1$ for all the experiments and the value of $b_2$ is tuned from 20 different values. Unless otherwise specified, all model parameters are chosen via 5-fold cross validation.

### 4.1 Benchmark Datasets

We use three datasets for our experiments. Details are given below:

**Landmine Detection**[3] consists of 19 tasks collected from different landmine fields. Each task is a binary classification problem: landmines $(+)$ or clutter $(-)$ and each example consists of 9 features extracted from radar images with four moment-based features, three correlation-based features, one energy ratio feature and a spatial variance feature. Landmine data is collected from two different terrains: tasks 1-10 are from highly foliated regions and tasks 11-19 are from desert regions, therefore tasks naturally form two clusters. Any hypothesis learned from a task should be able to utilize the information available from other tasks belonging to the same cluster.

**Spam Detection**[4] We use the dataset obtained from ECML PAKDD 2006 Discovery challenge for the spam detection task. We used the task B challenge dataset which consists of labeled training data from the inboxes of 15 users. We consider each user as a single task and the goal is to build a personalized spam filter for each user. Each task is a binary classification problem: spam $(+)$ or non-spam $(-)$ and each example consists of approximately $150K$ features representing term frequency of the word occurrences. Since some spam is universal to all users (e.g. financial scams), some messages might be useful to certain affinity groups, but spam to most others. Such adaptive behavior of user's interests and dis-interests can be modeled efficiently by utilizing the data from other users to learn per-user model parameters.

**Sentiment Analysis**[5] We evaluated our algorithm on product reviews from Amazon on a dataset containing reviews from 24 domains. We consider each domain as a binary classification task. Reviews with rating > 3 were labeled positive $(+)$, those with rating < 3 were labeled negative $(-)$, reviews with rating = 3 are discarded as the sentiments were ambiguous and hard to predict. Similar to the previous dataset, each example consists of approximately $350K$ features representing term frequency of the word occurrences.

We choose 3040 examples (160 training examples per task) for *landmine*, 1500 emails for *spam* (100 emails per user inbox) and 2400 reviews for *sentiment* (100 reviews per domain) for our experiments. We use the rest of the examples for test set. On average, each task in *landmine*, *spam*, *sentiment* has 509, 400 and 2000 examples respectively. Note that we intentionally kept the size of the training data small to drive the need for learning from other tasks, which diminishes as the training sets per task become large.

## 4.2 Results

To evaluate the performance of our proposed approach, we compare our proposed methods to 2 standard baselines. The first baseline selects the examples to query randomly (Random) and the second baseline chooses the examples via selective sampling independently for each task (Independent) [11]. We compare these baselines against two versions of our proposed algorithm 1 with different confidence measures for predictions from peer tasks: PEERsum where the confidence $\tilde{p}^{(t)}$ at round $t$ is computed by the weighted sum of the confidence of each task as shown originally in Algorithm 1 and PEERone where the confidence $\tilde{p}^{(t)}$ is set to the confidence of the most related task $k$ ($\hat{p}_k^{(t)}$), sampled from the probability distribution $\tau_{km}^{(t)}$, $m \in [K]$, $m \neq k$. The intuition is that, for multitask learning with many tasks [17], PEERone provides a faster computation without significantly compromising the performance of the learning algorithm. The task weights $\tau$ are computed based on the relationship between the tasks. As mentioned earlier, the $\tau$ values can be easily replaced by other functions based on the application at hand [6].

In addition to PEERsum and PEERone, we evaluated a method that queries the peer with the highest confidence, instead of the most related task as in PEERone, to provide the label. Since this method uses only local information for the task with highest confidence, it is not necessarily the best peer in hindsight, and the results are worse than or comparable (in some cases) to the Independent baseline. Hence, we do not report its results in our experiment.

Figure 2 shows the performance of the models during training. We measure the average rate of mistakes (cumulative measure), the number of label requests to the oracle and the number of peer query requests to evaluate the performance during the training time. From Figure 2 (top and middle), we can see that our proposed methods (PEERsum and PEERone) outperform both the baselines. Among the proposed methods, PEERsum outperforms PEERone as it uses the confidence from all the tasks (weighted by task relationship) to measure the final confidence. We notice that during the earlier part of the learning, all the methods issue more query to the oracle. After a few initial set of label requests, peer requests (dotted lines) steadily take over in our proposed methods. We can see three noticeable phases in our learning algorithm: initial label requests to the oracle, label requests to peers, and as task confidence grows, learning with less dependency on other tasks.

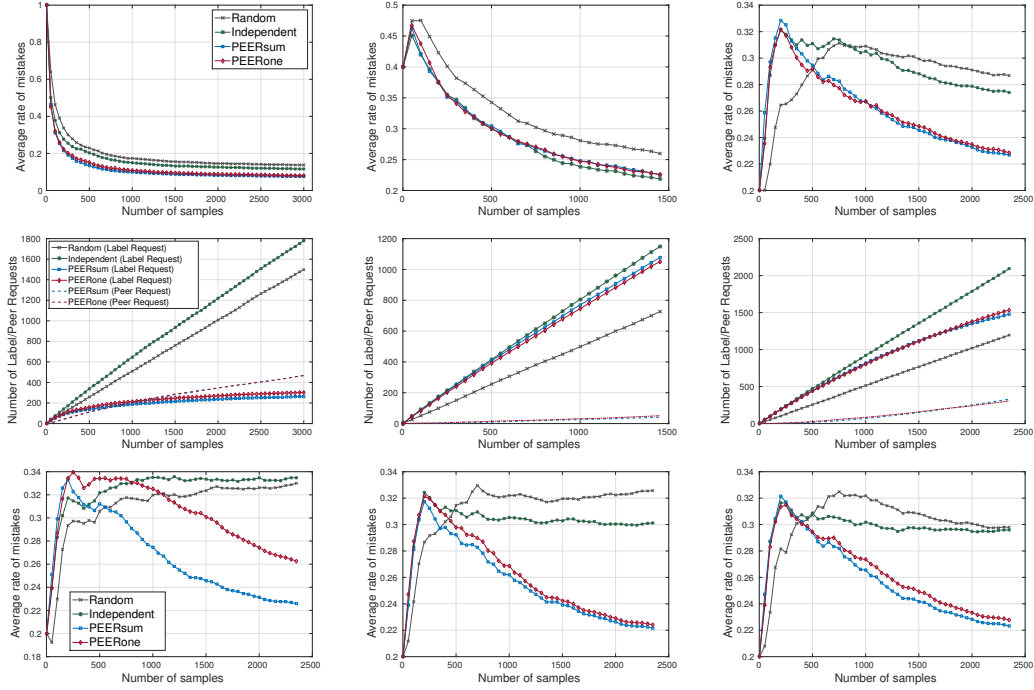

Figure 2: Average rate of mistakes vs. Number of examples calculated for compared models on the three datasets (top). Average number of label and peer requests on the three datasets (middle). Average rate of (training) mistakes vs. Number of examples with the query budget of $(10\%, 20\%, 30\%)$ of the total number of examples $T$ on *sentiment* (bottom). These plots are generated during the training.

In order to efficiently evaluate the proposed methods, we restrict the total number of label requests issued to the oracle during training, that is we give all the methods the same query budget: $(10\%, 20\%, 30\%)$ of the total number of examples $T$ on *sentiment* dataset. After the desired number of label requests to the oracle reached the said budget limit, the baseline methods predicts label for the new examples based on the earlier assistance from the oracle. On the other hand, our proposed methods continue to reduce the average mistake rate by requesting labels from peers. This shows the power of learning from peers when human expert assistance is expensive, scarce or unavailable.

Table 1 summarizes the performance of all the above algorithms on the test set for the three datasets. In addition to the average accuracy $ACC$ scores, we report the average total number of queries or label requests to the oracle ($\#Queries$) and the CPU time taken (seconds) for learning from $T$ examples (*Time*). From the table, it is evident that PEER* outperforms all the baselines in terms of both $ACC$ and $\#Queries$. In case of *landmine* and *sentiment*, we get a significant improvement in the test set accuracy while reducing the total number of label requests to the oracle. As in the training set results before, PEERsum performs slightly better than PEERone. Our methods perform slightly better than Independent in *spam*, we can see from Figure 2 (middle) for *spam* dataset, the number of peer queries are lower compared to that of the other datasets.

The results justify our claim that relying on assistance from peers in addition to human intervention leads to improved performance. Moreover, our algorithm consumes less or comparable CPU time than the baselines which take into account inter-task relationships and peer requests. Note that PEERone takes a little more training time than PEERsum. This is due to our implementation that takes more time in (MATLAB's) inbuilt sampler to draw the most related task. One may improve the sampling procedure to get better run time. However, the time spent on selecting the most related tasks is small compared to the other operations when dealing with many tasks.

Figure 3 (left) shows the average test set accuracy computed for 20 different values of $b_2$ for PEER* methods in *sentiment*. We set $b_1 = 1$. Each point in the plot corresponds to $ACC$ (y-axis) and $\#Queries$ (x-axis) computed for a specific value of $b_2$. We find the algorithm performs well for

Table 1: Average test accuracy on three datasets: means and standard errors over 10 random shuffles.

| Models | Landmine Detection | | | Spam Detection | | | Sentiment Analysis | | |
|---|---|---|---|---|---|---|---|---|---|
| | *ACC* | *#Queries* | *Time (s)* | *ACC* | *#Queries* | *Time (s)* | *ACC* | *#Queries* | *Time (s)* |
| Random | 0.8905 (0.007) | 1519.4 (31.9) | 0.38 | 0.8117 (0.021) | 753.4 (29.1) | 8 | 0.7443 (0.028) | 1221.8 (22.78) | 35.6 |
| Independent | 0.9040 (0.016) | 1802.8 (35.5) | 0.29 | 0.8309 (0.022) | 1186.6 (18.3) | 7.9 | 0.7522 (0.015) | 2137.6 (19.1) | 35.6 |
| PEERsum | **0.9403** (0.001) | 265.6 (18.7) | 0.38 | **0.8497** (0.007) | 1108.8 (32.1) | 8 | **0.8141** (0.001) | 1494.4 (68.59) | 36 |
| PEERone | 0.9377 (0.003) | 303 (17) | 1.01 | 0.8344 (0.018) | 1084.2 (24.2) | 8.3 | 0.8120 (0.01) | 1554.6 (92.2) | 36.3 |

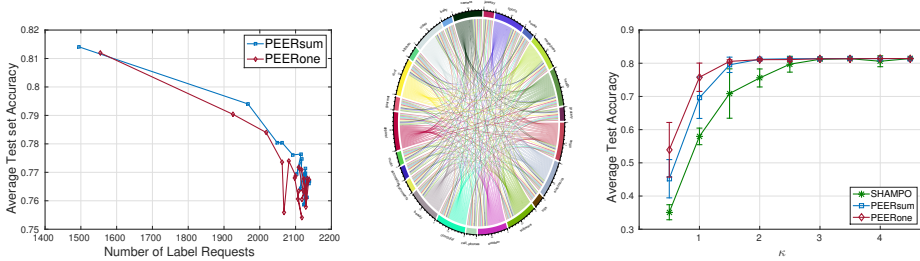

Figure 3: Average test set $ACC$ calculated for different values of $b_2$ (left). A visualization of the peer query requests among the tasks in *sentiment* learned by PEERone (middle) and comparison of proposed methods against SHAMPO in parallel setting. We report the average test set accuracy (right).

$b_2 > b_1$ and the small values of $b_2$. When we increase the value of $b_2$ to $\infty$, our algorithm reduces to the baseline (Independent), as all request are directed to the oracle instead of the peers.

Figure 3 (middle) shows the snapshot of the total number of peer requests between the tasks in *sentiment* at the end of the training of PEERone. Each edge says that there was one peer query request from a task/domain to another related task/domain (based on the task relationship matrix $\tau$). The edges with similar colors show the total number of peer requests from a task. It is evident from the figure that all the tasks are collaborative in terms of learning from each other.

Figure 3 (right) compares the PEER* implementation of Algorithm 2 in Appendix B against SHAMPO in terms of test set accuracy for *sentiment* dataset (See Supplementary material for more details on the Algorithm). The algorithm learns multiple tasks in parallel, where at most $\kappa$ out of $K$ label requests to the oracle are allowed at each round. While SHAMPO ignores the other tasks, our PEER* allows peer query to related tasks and thereby improves the overall performance. As we can see from the figure, when $\kappa$ is set to small values, PEER* performs significantly better than SHAMPO.

## 5 Conclusion

We proposed a novel online multitask learning algorithm that learns to perform each task jointly with learning inter-task relationships. The primary intuition we leveraged in this paper is that task performance can be improved both by querying external oracles and by querying peer tasks. The former incurs a cost or at least a query-budget bound, but the latter requires no human attention. Hence, our hypothesis was that with bounded queries to the human expert, additionally querying peers should improve task performance. Querying peers requires estimating the relation among tasks. The key idea is based on smoothing the loss function of each task w.r.t. a probabilistic distribution over all tasks, and adaptively refining such distribution over time. In addition to closed-form updating rules, we provided a theoretical bound on the expected number of mistakes. The effectiveness of our algorithm is empirically verified over three benchmark datasets where in all cases task accuracy improves both for PEERsum (sum of peer recommendations weighted by task similarity) and PEERone (peer recommendation from the most highly related task) over baselines such as assuming task independence.

## Footnotes

[1]We will consider a different online learning setting later in the supplementary section where we simultaneously receive $K$ examples at each round, one for each task $k$

[2]We also use the notation $\hat{p}_{kk} = \langle w_k^{(t-1)}, x^{(t)} \rangle$ and $\hat{p}_{km} = \langle w_m^{(t-1)}, x^{(t)} \rangle$

[3] http://www.ee.duke.edu/~lcarin/LandmineData.zip

[4] http://ecmlpkdd2006.org/challenge.html

[5] http://www.cs.jhu.edu/~mdredze/datasets/sentiment

[6] Our algorithm and theorem can be easily generalized to other types of functions on $\tau$

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
