[Supplementary Material]

# Supplementary Material for Active Learning from Peers

**Keerthiram Murugesan**    **Jaime Carbonell**
School of Computer Science
Carnegie Mellon University
Pittsburgh, PA 15213
{kmuruges,jgc}@cs.cmu.edu

## Appendix A   Proof of Theorem 1

*Proof.* We prove our theorem 1 for Algorithm 1 using the following lemma [1].

**Lemma 1.** *For a given example at round $t$, $\big((x^{(t)}, k), y^{(t)}\big)$, let $\alpha$ be some constant and $\gamma$ be the margin. Let $\{w_k^*\}_{k \in [K]}$ be any arbitrary vectors where $w_k^* \in \mathbb{R}^d$ and its hinge loss on the examples $\big(x^{(t)}, y^{(t)}\big)$ is given by $\ell_{kk}^{(t)*} = \big(\gamma - y^{(t)} \langle x^{(t)}, w_k^* \rangle\big)_+$. We have the following inequality:*

$$\alpha\gamma + |\hat{p}_k^{(t)}| \le \alpha\ell_{kk}^{(t)*} \le \alpha\ell_{kk}^{(t)*} + \frac{1}{2}\|\alpha w_k^* - w_k^{(t-1)}\|^2 - \frac{1}{2}\|\alpha w_k^* - w_k^{(t)}\|^2 + \frac{1}{2}\|w_k^{(t-1)} - w_k^{(t)}\|^2 + \frac{1}{2}\|w_k^{(t-1)} - w_k^{(t)}\|^2$$

*Proof.*

$$
\begin{aligned}
\gamma - \ell_{kk}^{(t)*} &= \gamma - \big(\gamma - y^{(t)} \langle x^{(t)}, w_k^* \rangle\big)_+ \\
&\le y^{(t)} \langle x^{(t)}, w_k^* \rangle \\
&= y^{(t)} \langle x^{(t)}, (w_k^* - w_k^{(t-1)} + w_k^{(t-1)}) \rangle \\
&= y^{(t)} \langle x^{(t)}, w_k^{(t-1)} \rangle + \frac{1}{2}\|w_k^* - w_k^{(t-1)}\|^2 - \frac{1}{2}\|w_k^* - w_k^{(t)}\|^2 + \frac{1}{2}\|w_k^{(t-1)} - w_k^{(t)}\|^2
\end{aligned}
$$

The above inequality holds for any $\gamma > 0$ and any arbitrary vector $w_k^*$, we replace $\gamma$ by $\alpha\gamma$ and $w_k^*$ by $\alpha w_k^*$ where $\alpha$ is some constant to be optimized. Since $y^{(t)} \langle x^{(t)}, w_k^{(t-1)} \rangle \le 0$ when we make a mistake at round $t$, we get our inequality by using the notation $\hat{p}_k^{(t)} = \langle x^{(t)}, w_k^{(t-1)} \rangle$.

$$\alpha\gamma + |\hat{p}_k^{(t)}| \le \alpha\ell_{kk}^{(t)*} + \frac{1}{2}\|\alpha w_k^* - w_k^{(t-1)}\|^2 - \frac{1}{2}\|\alpha w_k^* - w_k^{(t)}\|^2 + \frac{1}{2}\|w_k^{(t-1)} - w_k^{(t)}\|^2$$

Note that, for a task $m$ $(m \neq k)$, $y^{(t)} \langle x^{(t)}, w_m^{(t-1)} \rangle \le 0$ is not necessarily true and $\|w_m^{(t-1)} - w_m^{(t)}\|^2 = 0$ since $w_m^{(t)} = w_m^{(t-1)}$ at round $t$. □

To prove theorem 1, we bound the following two terms: $b_2\big(\alpha\gamma + |\hat{p}_k^{(t)}|\big) + \sum_{\substack{m \in [K] \\ m \neq k}} \tau_{km}^{(t)} |\hat{p}_m^{(t)}| \big(\alpha\gamma + |\hat{p}_k^{(t)}|\big)$ where $\hat{p}_m^{(t)} = \langle x^{(t)}, w_m^{(t-1)} \rangle$. Summing over $t$, we use $w_k^{(t)} = w_k^{(t-1)}$ when there is no mistake $(M^{(t)} = 0)$ and $\|w_k^{(t-1)} - w_k^{(t)}\|^2 \le X^2$ otherwise. We use $\sum_{t=1}^{T} [\frac{1}{2}\|\alpha w_k^* - w_k^{(t-1)}\|^2 - \frac{1}{2}\|\alpha w_k^* - w_k^{(t)}\|^2] = \frac{\alpha^2}{2}\|w_k^*\|^2$ and $w_k^{(0)} = 0$. Consider the $m^{th}$ task in the second term $(m \neq k)$, we have:

$$\sum_t M^{(t)} Z^{(t)} |\hat{p}_m^{(t)}| \big(\alpha\gamma + |\hat{p}_k^{(t)}| - \tfrac{X^2}{2}\big)$$

$$\leq \sum_t M^{(t)} Z^{(t)} \big[\alpha|\hat{p}_m^{(t)}| \ell_{kk}^{(t)*} + \tfrac{\alpha^2}{2}|\hat{p}_m^{(t)}| \|w_k^*\|^2\big]$$

$$\leq \sum_t M^{(t)} Z^{(t)} \big[\alpha\big(\gamma b_2 - |\hat{p}_m^{(t)}| y^{(t)} \langle x^{(t)}, w_k^*\rangle\big)_+ + \tfrac{\alpha^2 b_2}{2}\|w_k^*\|^2\big]$$

$$\leq \sum_t M^{(t)} Z^{(t)} \big[\alpha\big(\gamma b_2 - (y^{(t)}\langle x^{(t)}, w_m^*\rangle - \tfrac{1}{2}\|w_m^*\|^2)(y^{(t)}\langle x^{(t)}, w_k^{(t-1)}\rangle + \tfrac{1}{2}\|w_k^*\|^2 + \tfrac{X^2}{2}))_+ + \tfrac{\alpha^2 b_2}{2}\|w_k^*\|^2\big]$$

$$\leq \sum_t M^{(t)} Z^{(t)} \big[\alpha\big(\gamma b_2 - (y^{(t)}\langle x^{(t)}, w_m^*\rangle - \tfrac{1}{2}\|w_m^*\|^2)b_2\big)_+ + \tfrac{\alpha^2 b_2}{2}\|w_k^*\|^2\big]$$

$$\leq \sum_t M^{(t)} Z^{(t)} \big[b_2\big(\alpha\ell_{km}^{(t)*} + \tfrac{\alpha^2}{2}\|w_m^*\|^2 + \tfrac{\alpha^2}{2}\|w_k^*\|^2\big)\big]$$

where we have used the inequalities in Lemma 1 for both the tasks $k$ and $m$ ($m \neq k$). We set $b_2 = b_1 + \tfrac{1}{2}\|w_k^*\|^2 + \tfrac{X^2}{2}$. We choose $\alpha = (2b_1 + X^2)/2\gamma$. Now, combining both the terms, we have:

$$\sum_{t=1}^{T} M^{(t)} Z^{(t)}\bigg[\big(b_1 + |\hat{p}_k^{(t)}|\big)\big(b_2 + \sum_{\substack{m\in[K]\\m\neq k}} \tau_{km}^{(t)} |\hat{p}_m^{(t)}|\big)\bigg] \leq b_2 \bigg[\frac{(2b_1 + X^2)^2}{8\gamma^2}\big(\|w_k^*\|^2 + \sum_{\substack{m\in[K]\\m\neq k}} \tau_{km}^{(t)} \|w_m^*\|^2\big)$$

$$\frac{(2b_1 + X^2)}{2\gamma}\bigg(\sum_t M^{(t)} Z^{(t)} \ell_{kk}^{(t)*} + \sum_t \sum_{\substack{m\in[K]\\m\neq k}} \tau_{km}^{(t)} M^{(t)} Z^{(t)} \ell_{km}^{(t)*}\bigg)\bigg]$$

Taking expectation on both side and using $\tilde{L}_{kk} = \mathbb{E}\bigg[\sum_t M^{(t)} Z^{(t)} \ell_{kk}^{(t)*}\bigg]$ gives the desired result.

$\square$

## Appendix B    Learning Multiple Tasks in Parallel

In this section, we explore a related approach where multiple tasks are learned in parallel. In this setting, we assume that all tasks will be performed at each round. At time $t$, the $k^{th}$ task receives a training instance $x_k^{(t)}$, makes a prediction $\langle x_k^{(t)}, w_k^{(t)}\rangle$ and suffers a loss after $y^{(t)}$ is revealed. Unlike the problem setting in Algorithm 1, we are allowed to query the Oracle for at most $\kappa$ examples out of the $K$ examples received at round $t$ where $\kappa \leq K$. Our algorithm follows a perceptron-based update rule, as in Algorithm 1, in which the model is updated only when a task makes a mistake. The key idea is that the algorithm picks $\kappa$ examples from $K$ tasks that desperately need human assistance.

Most recently, Cohen et. al. [2] proposed a selective sampling-based approach called SHAMPO for this problem setting. Each task $k$ is chosen for label request from oracle with probability $\Pr(\mathcal{J}_i = k) \propto b_1/(b_1 + |\hat{p}_{kk}^{(t)}| - \min_{m=1}^{K} |\hat{p}_{mm}^{(t)}|), \forall i \in [\kappa]$ and we choose at most $\kappa$ tasks for label requests to oracle. We define $\mathcal{J} = \{\mathcal{J}_i : i \in [\kappa]\}$. Unlike in Algorithm 1, we perform a perceptron update when we make a mistake $M_k^{(t)} = \mathbb{I}(y_k^{(t)} \neq \hat{y}_k^{(t)})$ or when the example has less confidence $A_k^{(t)} = \mathbb{I}(0 < y_k^{(t)} \hat{p}_{kk}^{(t)} \leq \tfrac{\gamma}{2})$. In their proposed method, the examples from the other tasks $k \notin \mathcal{J}$ are not updated on this round. In addition, their methods doesn't take into account the relationship between the tasks. Our learning procedure from Algorithm 1 provides a natural way to extend their method to learn from peers and to utilize the relationship between the tasks efficiently.

The pseudo-code is in Algorithm 2. Lines $(9 - 12)$ is similar to their proposed learning framework. We add the lines $(15 - 18)$ to query the peers for labels for the tasks that are not selected at this

---

**Algorithm 2:** Learning Multiple Tasks in Parallel from Peers

---
**Input :** $b_1 > 0, b_2 > 0$ s.t., $b_2 \geq b_1, \lambda > 0, \gamma > 0$ Number of rounds $T$

1  *Initialize* $w_m^{(0)} = \mathbf{0} \;\forall m \in [K], \boldsymbol{\tau}^{(0)}$.

2  **for** $t = 1 \ldots T$ **do**

3     *Receive* $K$ examples: $\{x_k^{(t)} : k \in [K]\}$

4     *Compute* $\hat{p}_{kk}^{(t)} = \langle x_k^{(t)}, w_k^{(t-1)} \rangle, k \in [K]$

5     *Predict* $K$ labels: $\hat{y}_k^{(t)} = sign(\hat{p}_{kk}^{(t)}), k \in [K]$

6     *Draw* $\kappa$ tasks for query with probability

      $\Pr(\mathcal{J}_i = k) \propto b_1/(b_1 + |\hat{p}_{kk}^{(t)}| - \min_{m=1}^{K} |\hat{p}_{mm}^{(t)}|), \forall i \in [\kappa]$

7     **for** $k = 1 \ldots [K]$ **do**

8       **if** $k \in \mathcal{J}$ **then**

9         *Query* true label $y_k^{(t)}$

10         Set $M_k^{(t)} = 1$ if $\hat{y}_k^{(t)} \neq y_k^{(t)}$

11         Set $A_k^{(t)} = 1$ if $0 < y_k^{(t)} \hat{p}_{kk}^{(t)} \leq \frac{\gamma}{2}$

12         *Update* $w_k^{(t)} = w_k^{(t-1)} + (M_k^{(t)} + A_k^{(t)}) y_k^{(t)} x_k^{(t)}$.

13         *Update* $\tau^{(t)}$ as in Equation 1.

14       **else**

15         *Compute* $\hat{p}_{km}^{(t)} = \langle x_k^{(t)}, w_m^{(t-1)} \rangle \;\forall m \neq k, m \in [K]$

16         *Compute* $\tilde{p}_k^{(t)} = \sum_{m \neq k, m \in [K]} \tau_{km}^{(t-1)} \hat{p}_{km}^{(t)}$ and $\tilde{y}_k^{(t)} = sign(\tilde{p}_k^{(t)})$

17         *Draw* a Bernoulli random variable $\tilde{Z}_k^{(t)}$ with probability $\frac{|\tilde{p}_k^{(t)}|}{b_2 + |\tilde{p}_k^{(t)}|}$

18         *Update* $w_k^{(t)} = w_k^{(t-1)} + \tilde{Z}_k^{(t)} \tilde{y}_j^{(t)} x_k^{(t)}$

19       **end**

20     **end**

21 **end**

---

round $k \notin \mathcal{J}$ and the line 13 to incorporate the relationship between the tasks when the true labels are available. We give the (expected) mistake bound for the Algorithm 2 in Theorem 2.

**Theorem 2.** $\forall k \in [K]$, *let* $S_k = \left\{ \left( x_k^{(t)}, y_k^{(t)} \right) \right\}_{t=1}^{T}$ *be a sequence of* $T$ *examples for the* $k^{th}$ *task where* $x_k^{(t)} \in \mathbb{R}^d$, $y_k^{(t)} \in \{-1, +1\}$ *and* $\|x_k^{(t)}\|_2 \leq R, \forall t \in [T]$. *Let* $M_k^{(t)} = \mathbb{I}(y_k^{(t)} \neq \hat{y}_k^{(t)})$ *and* $A_k^{(t)} = \mathbb{I}(0 < y_k^{(t)} \hat{p}_{kk}^{(t)} \leq \frac{\gamma}{2})$.

*If* $\{S_k\}_{k \in [K]}$ *is presented to Algorithm 2 with* $b_1 > 0 \; (b_1 \geq \gamma)$ *and* $b_2 > 0 \; (b_2 \geq b_1)$, *then* $\forall t \geq 1$ *and* $\gamma > 0$ *we have*

$$\mathbb{E}\left[\sum_{k \in [K]} \sum_{t \in [T]} M_k^{(t)}\right] \leq \frac{b_2 K}{\gamma}\left[\frac{(2b_1 + X^2)^2}{8 b_1 \gamma}\left(\|w_k^*\|^2 + \max_{m \in [K], m \neq k} \|w_m^*\|^2\right)\right.$$
$$\left. + \left(1 + \frac{X^2}{2b_1}\right)\left(\tilde{L}_{kk} + \max_{m \in [K], m \neq k} \tilde{L}_{km}\right)\right] + \left(\frac{\gamma}{b_1} - 1\right)\mathbb{E}\left[\sum_{k \in [K]} \sum_{t \in [T]} A_k^{(t)}\right]$$

The proof is straight-forward and follows directly from the proof of Theorem 1 and Theorem 1 in [2]. The first two terms in the bound is same as in Theorem 1. The last term in the bound accounts for the aggressiveness of the algorithm. The intuition is that when we set $b_1$ to the margin (i.e., $b_1 = \gamma$), the last term will become 0 and the bound reduces to the one given in Theorem 1. When $b_1 > \gamma$, the aggressive term in the bound reduces the expected number of the mistakes made by Algorithm 2 and increases the expected number of label requests to the peers and eventually to the oracle.