[Reviews · NeurIPS 2017]

Reviewer 1



The authors propose an extension of the active learning framework to address two key issues: active learning of multiple related tasks, and using the 'peers' as an additional oracle to reduce the cost of querying the user all the time. The experimental evaluation of the proposed learning algorithms is performed on several benchmark datasets, and in this context, the authors proved their claims. The main novelty lies in the extension of the SHAMPO algorithm so that it can leverage the information from multiple tasks, and also peer-oracles (using the confidence in the predictions), to reduce the annotation efforts by the user. A sound theoretical formulation to incorporate these two is proposed. Experimental results show the validity of the proposed. The authors did not comment on the cost of using the peers as oracles, but assumed their availability. This should be detailed more. Also, it is difficult to assess the model's performance fully compared to the SHAMPO algorithm as its results are missing in Table 1. The accuracy difference between peer and one version of the model is marginal. What is the benefit of the former? Overall, this is a nice paper with a good mix of theory and empirical analysis. Lots of typos in the text that should be corrected.

Reviewer 2



This paper proposes an online multitask learning algorithm. It claims that it can jointly learn the parameters and task-relatedness. However, there is a lack of comparison with other related work that can also do the joint work. Thus the major contribution of this work is not clear. I would like to see a clear comparison with other papers that can learn the task relatedness. Minor typos: line 242 algorithm ?

Reviewer 3



This paper proposes a novel active learning approach in an online multitask setting, where the algorithm can optionally query labels from peers or oracles. For a specific task model, its parameters are estimated by utilizing information from peer models with the joint learning inter-task relationships. Experiments are performed on three benchmark datasets, and results validated the superiority of the proposed method over baselines. Theoretical analysis on the error bound is present. However, many key steps are directly following existing studies. The reference is insufficient. More related studies should be cited. The presentation should be significantly improved.